# Clinical and Cytogenetic Impact of Maternal Balanced Double Translocation: A Familial Case of 15q11.2 Microduplication and Microdeletion Syndromes with Genetic Counselling Implications

**DOI:** 10.3390/genes15121546

**Published:** 2024-11-29

**Authors:** Daniela Koeller R. Vieira, Ingrid Bendas Feres Lima, Carla Rosenberg, Carlos Roberto da Fonseca, Leonardo Henrique Ferreira Gomes, Letícia da Cunha Guida, Patrícia Camacho Mazzonetto, Juan Llerena, Elenice Ferreira Bastos

**Affiliations:** 1Centro de Genética Médica, Instituto Nacional da Saúde da Mulher, da Criança e do Adolescente Fernandes Figueira–Fundação Oswaldo Cruz, Rio de Janeiro 22250-020, Brazil; juan.llerena@fiocruz.br; 2Secretaria Municipal de Saúde de Angra dos Reis, Angra dos Reis 23906-010, Brazil; 3Laboratório de Citogenética Clínica, Centro de Genética Médica, Instituto Nacional da Saúde da Mulher, da Criança e do Adolescente Fernandes Figueira–Fundação Oswaldo Cruz, Rio de Janeiro 22250-020, Brazil; ingridlima@aluno.fiocruz.br (I.B.F.L.); carlos.rfonseca@fiocruz.br (C.R.d.F.); 4The Human Genome and Stem Cell Research Center, Department of Genetics and Evolutionary Biology, Institute of Biosciences, University of São Paulo, São Paulo 05508-220, Brazil; carlarosenberg2@gmail.com (C.R.); patricia.mazzonetto@gmail.com (P.C.M.); 5Diagnósticos da América S.A., DASA, São Paulo 06455-010, Brazil; 6Departamento de Pesquisa Clínica, Instituto Nacional da Saúde da Mulher, da Criança e do Adolescente Fernandes Figueira–Fundação Oswaldo Cruz, Rio de Janeiro 22250-020, Brazil; leonardo.henrique@fiocruz.br (L.H.F.G.); leticia.guida@fiocruz.br (L.d.C.G.)

**Keywords:** double balanced translocation, genetic couseling, FISH, LP-WGS

## Abstract

Background: Balanced chromosomal translocations occur in approximately 0.16 to 0.20% of live births. While most carriers are phenotypically normal, they are at risk of generating unbalanced gametes during meiosis, leading to genetic anomalies such as aneuploidies, deletions, duplications, and gene disruptions. These anomalies can result in spontaneous abortions or congenital anomalies, including neurodevelopmental disorders. Complex chromosomal rearrangements (CCRs) involving more than two chromosomes are rare but further increase the probability of producing unbalanced gametes. Neurodevelopmental disorders such as Angelman syndrome (AS) and duplication 15q11q13 syndrome (Dup15q) are associated with such chromosomal abnormalities. Methods: This study describes a family with a de novo maternal balanced double translocation involving chromosomes 13, 19, and 15, resulting in two offspring with unbalanced chromosomal abnormalities. Cytogenetic evaluations were performed using GTG banding, fluorescence in situ hybridization (FISH), and low-pass whole-genome sequencing (LP-WGS). Methylation analysis was conducted using methylation-sensitive high-resolution melting (MS-HRM) to diagnose Angelman syndrome. Results: The cytogenetic and molecular analyses identified an 8.9 Mb duplication in 15q11.2q13.3 in one child, and an 8.9 Mb deletion in the same region in the second child. Both abnormalities affected critical neurodevelopmental genes, such as *SNRPN*. FISH and MS-HRM confirmed the chromosomal imbalances and the diagnosis of Angelman syndrome in the second child. The maternal balanced translocation was found to be cryptic, contributing to the complex inheritance pattern. Conclusion: This case highlights the importance of using multiple genetic platforms to uncover complex chromosomal rearrangements and their impact on neurodevelopmental disorders. The findings underscore the need for thorough genetic counseling, especially in families with such rare chromosomal alterations, to manage reproductive outcomes and neurodevelopmental risks.

## 1. Introduction

Balanced chromosomal translocations occur in approximately 0.16 to 0.20% of live births. Most cases are associated with a normal phenotype; however, carriers of balanced translocations are at increased risk of producing unbalanced gametes, leading to various genetic anomalies such as large segmental aneuploidies, submicroscopic deletions and/or duplications, gene disruptions, and others [1]. These anomalies can result in spontaneous abortions and/or children with congenital anomalies associated or not with neurodevelopmental disorders [2]. Complex chromosomal rearrangements involving more than two chromosomes with segment exchanges are rare, mostly occurring de novo. The occurrence of two or three independent translocations is extremely rare, and when they do occur, they significantly increase the probability of having children with segmental aneuploidies [2,3].

Neurodevelopmental disorders and multiple congenital anomalies are major drivers of chromosomal structural abnormalities. Depending on their size, these alterations can be detected by karyotype, fluorescence in situ hybridization (FISH), and chromosomal microarray/low-pass whole-genome sequencing (LP-WGS), but only the first two can detect balanced translocations [3,4]. Complex chromosome rearrangements (CCR) are translocations that involve two or more chromosomes at more than two breakpoints [5,6]. In general, carriers of balanced reciprocal translocations are phenotypically normal; however, their offspring may present with symptoms when they inherit unbalanced gametes during meiosis, known as copy number variations (CNVs) [3,4,5,6]. In a reciprocal translocation, CNVs may result from non-allelic exchanges between non-homologous chromosomes (non-allelic homologous recombination, NAHR), driven by repeated DNA sequences known as low-copy repeats (LCRs) [3,7]. This mechanism occurs during meiosis in the quadrivalent configuration when normal and balanced chromosomes participate in NAHR [8,9]. During meiotic segregation, different chromosomal combinations can be generated, giving carriers of reciprocal translocations a high chance of producing unbalanced gametes [6,10]. Proximal 15q is a cluster of LCRs susceptible to recombination events that generate deletions or duplications associated with Angelman/Prader–Willi and 15q11q13 microduplication syndromes, respectively [9].

Angelman syndrome (AS) and duplication of 15q11.1-13.1 (Dup15q syndrome) are both rare genetic disorders affecting different regions of chromosome 15. AS, occurring in 1 in 10,000 to 30,000 individuals, is characterized by severe developmental delays, movement and balance disorders, hypotonia, and unique behavioral traits, like a happy disposition and frequent laughter. AS is primarily caused by deletions or imprinting defects in the 15q11-q13 region or mutations in the *UBE3A* gene. Treatment is multidisciplinary, focusing on managing symptoms and promoting the child’s development [10]. Dup15q syndrome, affecting 1 in 30,000 to 1 in 60,000 children, involves an extra copy of part of chromosome 15, leading to hypotonia, epilepsy, cognitive and motor delays, autism, and distinct facial features. Treatment includes medications for various symptoms, along with occupational, physical, and behavioral therapies to enhance patients’ quality of life and functionality [11].

Here, we describe a family with a de novo maternal balanced double translocation involving chromosomes 13, 19, and 15 [46,XX, t(13:19)(q22;p13.1).ish t(13:15)(q12;q12)] and two offspring with different unbalanced chromosomal abnormalities, presenting global developmental delay and congenital anomalies associated with Angelman syndrome and 15q11q13 microduplication syndrome.

## 2. Materials and Methods

### 2.1. Sample Collection and DNA Isolation

Blood samples were collected from 2 individuals that belong to a previous study [12] that were analyzed in a coorte of 1363 patients with unexplained neurodevelopmental delay/intellectual disability, autism spectrum disorder, and/or multiple congenital anomalies. They were patients from Angra dos Reis city (South zone in Rio de Janeiro State, Brazil). The patients were enrolled in Fernandes Figueira National Institute (IFF/FIOCRUZ). As a control, blood samples from healthy individuals were collected. DNA extraction was performed using the DNAeasy Blood & Tissue Kit obtained from Qiagen (Valencia, CA) following instructions provided by the standard kit protocol. The DNA quality was assessed with a NanoDrop Spectrophotometer (ThermoFisher Scientific, Wilmington, NC, USA). The DNA purity was also evaluated through the wavelengths 260/280 and 260/230, avoiding contaminants.

The research was carried out in accordance with the Declaration of Helsinki and received approval from the Institutional Review Board of the University of São Paulo (CAAE: 53093821.3.0000.5464). Informed consent was secured from all guardians of the participants involved in the study. Additionally, written consent was obtained from the guardians for the publication of this document.

### 2.2. LP-WGS

LP-WGS were performed to 1x coverage using the Illumina next-generation sequencing (NGS) platform [13]. Data analyses were performed using NxClinical software 6.0,(BioDiscovery, San Diego, CA, USA).

### 2.3. Cytogenetic and FISH Analysis

The cytogenetic study was performed according to a standard protocol: a heparinized peripheral blood sample was collected from patients and all family members. Lymphocyte cultures were performed for 72 h, stimulated with phytohemagglutinin (PHA), and maintained in RPMI 1640 medium and 20% fetal bovine serum. The culture was terminated after 1 h in the presence of colchicine (16 μg/mL) and successive fixations in methanol/acetic acid (3:1). The slides were prepared and stained with GTG banding (trypsin/giemsa). A minimum of 20 metaphases were analyzed at a resolution of 450/550 bands per haploid set; the images were captured by the automated cell imaging system CytoVision^®^, and the karyotypes were described according to the International System for Human Cytogenetic Nomenclature (2020).

The FISH analyses in the samples of proband and her sister were performed using the Prader–Willi/Angelman region probe (15q11.2—*SNRPN*) marked in red and the subtelomere region (15qter) marked in green.

The sample of proband’s mother was investigated by FISH using the Prader–Willi/Angelman region probe together with the chromosome 13 centromeric probe (D13Z1, marked in green). All probes used were obtained commercially (Cytocell, Inc., Milton, Cambridge, UK).

For both, the slides were treated with 2X SSC for 30 min and dehydrated in 70%, 85%, and 100% alcohol for 2 min each. The probes were applied to the slide, mounted with a coverslip, and co-denatured for 7 min on a hot plate at 72 °C. After staying overnight, the coverslips were removed, and post-hybridization washes were performed in 0.4 × SSC solution at 72 °C for 1 min, followed by washing in 2 × SCC 0.05% Tween for 2 min. After DAPI staining, the material was covered with a coverslip, observed under an epifluorescent microscope, and images were captured by a Cytovision Inc. imaging system Cytovision Inc., Auburn, AL, USA).

### 2.4. Bisulfite DNA Treatment

The genomic DNA input was 250 ng/μL, to be modified with sodium bisulfite using the EZ DNA^™^ methylation kit (Zymo Research, Tustin, CA, USA). The final protocol step was to elute in 10 μL of nuclease-free water according to the manufacturer’s instructions. The modified DNA was quantified and was assessed with a NanoDrop spectrophotometer (ThermoFisher Scientific, Wilmington, NC, USA).

### 2.5. Methylation-Sensitive High-Resolution Melting

Methylation-sensitive high-resolution melting (MS-HRM) was performed on the 7500 Fast Real-Time PCR System mix (ThermoFisher Scientific, Wilmington, NC, USA) according to the literature [14,15,16]. Each sample was analyzed in triplicate for MS-HRM. Primers were designed according to the principles outlined by Wojdacz and Hansen (2006) [17]. The primers used to amplify bisulfite-treated DNA and unmodified genomic DNA were PWS_F (5′-GGATTTTTGTATTGCGGTAAATAAG-3′) and PWS_R (5′-CAACTAACCTTACCCACTCCATC-3′). The amplicon size of the methylated maternal allele and the nonmethylated paternal allele is 187 bp. The PCR reaction was performed in 200 μL PCR tubes with a final volume of 10 μL, containing 200 nmol/L of each primer, 5 μL of HRM-Master Mix (ThermoFisher Scientific, Wilmington, NC, USA), and 10 ng of bisulfite-treated DNA assessed with a NanoDrop spectrophotometer. The initial denaturation (95 °C, 15 min) was followed by 45 cycles for MS-HRM of 15 s at 95 °C, 1 min at 60 °C, and a HRM step from 60 °C to 90 °C, rising at 0.2 °C per s and holding for 1 s after each stepwise increment. The amplification products from unmethylated and methylated alleles had melting temperatures of 78 °C and 83 °C, respectively.

### 2.6. Methodology of Bibliographic Research

To explore, in greater detail, the genetic landscape of the chromosomal anomalies observed in this family, an extensive literature review was conducted. The search was performed in the PubMed database, using the following keywords: “balanced double translocation”, “complex rearrangements”, “15q11.2 duplication”, “Angelman syndrome”, and “15q11.2 deletion”. These terms were selected with the aim of capturing relevant studies related to both chromosomal structural rearrangements and their phenotypic consequences.

Studies published in English, without time restrictions, were included to ensure a comprehensive review of current and past knowledge. Additionally, to verify the correlation between the findings from the literature review, pathogenic variants already described in genomic databases such as Genome Browser *hg38* (GRCh38) and v11.28 of the Database of Chromosomal Imbalance and Phenotype in Humans using Ensembl Resources (DECIPHER) were consulted. These tools allowed for the comparison of the duplicated and deleted variants in 15q11.2 with previously identified and documented variants, facilitating the assessment of their clinical significance.

The articles selected for review had to address balanced double translocations and complex chromosomal rearrangements; phenotypic implications of duplication and/or deletion in the 15q11.2 segment, especially in conditions such as Angelman syndrome; and clinical studies or systematic reviews that discussed the correlation between chromosomal alterations and clinical outcomes in families carrying these anomalies.

Articles that did not focus on the chromosomal regions of interest or involved rearrangements unrelated to the 15q11.2 duplication or deletion were excluded. The search was conducted with specific filters to limit the selection to relevant clinical studies, reviews, and case reports. All retrieved articles were initially evaluated by title and abstract, and those meeting the inclusion criteria were read in full for detailed analysis. Key information about the clinical implications of chromosomal rearrangements and their phenotypic consequences was extracted to form the theoretical basis of the genetic analysis of the family under study.

The findings obtained through the genomic databases were cross-referenced with the bibliographic data to identify clinically relevant genetic variants and associate these variants with the clinical profile presented by the family, highlighting both duplication and deletion cases in the 15q11.2 region.

## 3. Results

### 3.1. Case Report

Prenatal care was monitored throughout the pregnancy, with negative serologies (STORCH), and the mother reported insulin use close to delivery due to gestational diabetes. The baby was delivered by cesarean section (due to a previous cesarean) at 37 weeks and 5 days of gestation, with an APGAR score of 8/9, and a birth weight of 2505 g, being classified as small for gestational age. There were no neonatal complications, and neonatal screening was normal.

The patient is a girl who was referred at 4 years of age due to global developmental delay. She is the first of two daughters of a non-consanguineous couple, with a healthy mother and a father with vitiligo. The pregnancy was monitored with prenatal care, and the patient was delivered by cesarean section at 39 weeks and 2 days due to oligohydramnios and acute fetal distress. The APGAR score was 9/9. Her birth weight was 2390 g, with a length of 46 cm and a head circumference of 32 cm, being classified as small for gestational age. There were no perinatal complications, and neonatal screening was normal.

In the first few months of life, the patient had difficulty sucking, sat up at 18 months, and began walking and speaking her first words at 30 months. At 4 years old, an abdominal ultrasound revealed an image in the mesogastrium, anterior to the lumbar spine, resembling a kidney with typical architecture but not related to the right and left kidneys, suggesting a supernumerary ectopic kidney (48 × 16 × 32 mm). Auditory evoked potentials and echocardiogram were normal, and there was no history of previous hospitalizations.

At 7 years old, the patient still speaks only isolated words, can feed herself using a spoon, and has daytime bladder control, but requires nighttime diaper use. She is enrolled in a school with special education needs. Physical examination revealed small, pointed ears with a depression in the left ear, ocular hypotelorism, a small mouth with downturned corners, deep-set eyes, short stature, and low weight (107 cm and 15,100 g, both below the 3rd percentile).

### 3.2. Molecular Analysis

Low-pass whole-genome sequencing (LP-WGS) analysis identified a pathogenic 8.9 Mb duplication on chromosome 15q11.2q13.3 in the proband, documented as seq [GRCh38] 15q11.2q13.3(22189863_31089864)x3 (Figure 1A) and a pathogenic 8.9 Mb deletion on chromosome 15q11.2q13.3 in her sister (Figure 1B).

Methylation analysis of the *SNURPN/SNURF* gene in the proband’s sibling, conducted via MS-HRM, confirmed an exclusively paternal methylation pattern, supporting an Angelman syndrome diagnosis (Figure 2).

### 3.3. Analysis by Classical and Molecular Cytogenetic (FISH)

GTG banding analysis was carried out and showed an apparently normal karyotype in the proband (46,XX—Figure 3A), while his sister had an apparently balanced translocation between chromosomes 13 and 19, with apparently normal chromosome 15 by the resolution investigated, karyotype: 46,XX, t(13;19)(q22;p13.1), Figure 3B. FISH analysis with Prader–Willi Angelman probe *SNRPN* demonstrated the duplication of the 15q11.2 region in the proband (Figure 3A) and deletion at 15q11.2 in her sister (Figure 3B).

GTG banding analysis of the mother showed the same balanced translocation t(13;19) with apparently normal chromosome 15, as observed in one of the daughters (Figure 4A,B). FISH analysis using the Prader–Willi/Angelman probes (15q11.2 in red and 15qter in green) and the probe for the centromeric region of chromosome 13 (green) identified another apparently balanced translocation (cryptic) between chromosomes 13 and 15 not involved in the initial translocation (Figure 4C).

## 4. Discussion

The analysis of this family (Figure 5) revealed complex genetic findings, beginning with two siblings exhibiting distinct chromosomal abnormalities and related clinical phenotypes. The proband presented with a pathogenic 8.9 Mb duplication on chromosome 15q11.2q13.3, which includes regions critical for neurodevelopment, notably the *SNRPN* gene, a locus implicated in Prader–Willi syndrome (PWS) and Angelman syndrome (AS). The younger sibling exhibited a deletion in this same region, corresponding with typical Angelman syndrome phenotypes, including developmental delay, axial hypotonia, and seizures.

The findings from the FISH analysis provide key insights into the chromosomal rearrangements observed in both the proband and her sister. The proband’s 15q11.2 region triplication and her sister’s deletion confirm the genetic basis of their respective clinical phenotypes. These structural variations, affecting the same chromosomal region yet in opposing manners (duplication versus deletion), underscore the diverse phenotypic manifestations associated with 15q11.2q13.3 abnormalities, including 15q11q13 microduplication syndrome in the proband and Angelman syndrome in her sister.

The *SNRPN* gene is involved in mRNA processing, which is vital for various neuronal activities. Located on chromosome 15, the *SNRPN* locus is essential for epigenetic control, where disruptions contribute to neurodevelopmental syndromes. In PWS, there are distinct cognitive and behavioral manifestations alongside hypothalamic dysregulation, affecting growth and appetite. In AS, which results from maternal deletions or mutations within this region, symptoms include severe developmental delay, motor dysfunction, and seizures. These neurodevelopmental disorders reflect the critical influence of SNRPN on brain development and behavior [17,18].

Furthermore, deletions in the *SNORD116* region, associated with *SNRPN*, result in metabolic and behavioral disruptions in PWS due to appetite and growth dysregulation. Studies on PWS and AS patients, along with mouse models, have shown that genetic disruptions in these areas lead to circadian rhythm gene desynchronization, likely impacting neurodevelopment and behavior [18].

Low-pass whole-genome sequencing (LP-WGS) identifed a pathogenic 8.9 Mb duplication on chromosome 15q11.2q13.3 in the proband, documented as seq[GRCh38] 15q11.2q13.3(22189863_31089864)x3. This region includes *SNRPN* and other loci implicated in imprinting disorders. FISH analysis validated the 15q11.2 duplication in the proband (Figure 1), supporting the LP-WGS findings.

For the proband’s younger sister, who exhibited developmental delay, hypotonia, and neurological symptoms, cytogenetic analysis initially identified a balanced translocation between chromosomes 13 and 19 (46,XX, t(13;19)(q22;p13.1)). LP-WGS later identified a pathogenic 8.9 Mb deletion on chromosome 15q11.2q13.3, confirmed by FISH (Figure 3), consistent with Angelman syndrome.

In light of these results, we investigated the chromosomal makeup of the parents, initially by GTG banding. The father’s karyotype was normal, but in the mother, we identified a balanced translocation between one of chromosomes 13 and chromosome 19 46,XX, t(13;19(q22;p13.1). This finding did not explain the occurrence of the alterations identified in the daughters. Considering a possible complex rearrangement involving 13q and 15q, we continued the mother’s investigation by performing the FISH technique using probes that recognize these regions. Surprisingly, we found that there was another translocation involving the regions of chromosomes 15q11.2 and 13q11.1 (Figure 4C) that were apparently normal by GTG banding, characterizing a balanced double translocation in the mother (Figure 4). This rearrangement was considered cryptic since it was undetectable by classical cytogenetics.

Other relatives, including the father, brother, maternal aunt, and grandparents were evaluated by GTG banding analysis and the FISH technique, all of them presenting normal results, suggesting that this cryptic translocation arose “de novo” in the mother. The mother’s karyotype was classified as: 46,XX, t(13;19)(q22;p13.1).ish t(13;15)(q11.1;q11.2) “de novo”.

Although a rare event, this double translocation did not cause phenotypic effects in the mother, since they were balanced exchanges between the chromosomes. Carriers of balanced translocations generally do not present clinical alterations, but they are at greater risk of infertility, spontaneous abortions and the production of unbalanced gametes [2]. In this case, the segregation of the chromosomes during maternal meiosis resulted in the absence of the chromosome 13 derivative in the proband, leading to monosomy of the 13q11.1 region and a duplication of 15q11.2 region. The proband’s sister carries a balanced translocation 13;19 mat, and a loss of the 15q11.2 region characterizing Angelman syndrome.

Methylation analysis of the *SNURPN/SNURF* gene in the proband’s sibling using MS-HRM confirmed an exclusively paternal methylation pattern, validating an Angelman syndrome diagnosis (Figure 2). MS-HRM proved to be a rapid and reliable method to detect the loss of the maternal allele, a hallmark of AS.

A systematic PubMed search using Boolean operators provided targeted insights into the relationship between 15q11.2 rearrangements and complex chromosomal conditions. For the query “(15q11.2 duplication) AND (complex rearrangements)”, 10 results were retrieved, of which only 3 were relevant. The search terms “(15q11.2 duplication) AND (balanced double translocation)” and “(15q11.2 deletion) AND (balanced double translocation)” returned no results, while “(15q11.2 deletion) AND (complex rearrangements)” yielded 6 results, with only 1 being relevant. For “(Angelman syndrome) AND (complex rearrangements)”, 17 results were obtained, of which 5 were considered relevant, but no results were found with the combination “(Angelman syndrome) AND (balanced double translocation)”.

Given the complexity of the genetic findings in this family, comprehensive genetic counseling plays a pivotal role in managing the reproductive and familial implications. While the recurrence risk for the translocation or related chromosomal imbalances remains low, the variability in phenotypic expression among offspring highlights the need for careful prenatal screening in future pregnancies. Techniques such as karyotyping and LP-WGS may help identify chromosomal abnormalities early, allowing for informed reproductive decisions [19]. Counseling should also include discussions about the potential risks of passing on either balanced translocations or chromosomal imbalances, as even balanced carriers may experience reproductive challenges. Early genetic counseling, combined with appropriate prenatal testing, will aid in providing the family with the necessary tools to anticipate and manage future reproductive outcomes [20].

Moreover, the incorporation of 100× whole-genome sequencing (WGS) offers deeper genome coverage, enhancing the detection of single-nucleotide variants (SNVs), insertions, deletions, and copy number variations (CNVs). Long-read sequencing technologies, such as those provided by Pacific Biosciences (PacBio) and Oxford Nanopore, enable the sequencing of longer DNA fragments. This capability is particularly advantageous for capturing repetitive regions and large structural variants often missed by traditional approaches. Research indicates that these technologies can significantly improve the accuracy of variant detection and facilitate the assembly of complex genomic regions [21,22].

As the cost-effectiveness of these sequencing technologies continues to improve, clinical genetic diagnostic labs can adopt WGS as a standard tool. This not only provides comprehensive genomic information to inform clinical decisions and enhance patient outcomes but also allows for the uncovering of previously undiagnosed genetic conditions, enabling tailored treatment strategies. Integrating advanced sequencing technologies into clinical practice not only enhances diagnostic capabilities but also reflects the ongoing evolution of precision medicine, ultimately benefiting patient care and research initiatives [22,23].

## 5. Conclusions

This case exemplifies the significant impact that complex chromosomal rearrangements can have on neurodevelopment, highlighting the critical need for a multidisciplinary diagnostic approach in the investigation of such anomalies. The use of complementary cytogenetic and molecular techniques has enabled the identification of duplications and deletions that significantly influence clinical outcomes. Specifically, the presence of the 15q11.2q13.3 duplication and deletion in siblings highlights the potential for balanced translocations to result in diverse and severe neurodevelopmental disorders. The use of the FISH technique was fundamental for the identification of the maternal cryptic translocation, leading to a better understanding of the mechanism involved in the origin of the alterations in the sisters, as a consequence of the segregation of these chromosomes in maternal meiosis. The occurrence of this maternal double translocation, an extremely rare event, underlines the importance of more comprehensive genetic counseling for families affected by rare chromosomal anomalies. This counseling should address both the reproductive implications and the lifelong management of the physical and neurodevelopmental challenges that can arise due to these genetic rearrangements.

## Figures and Tables

**Figure 1 genes-15-01546-f001:**
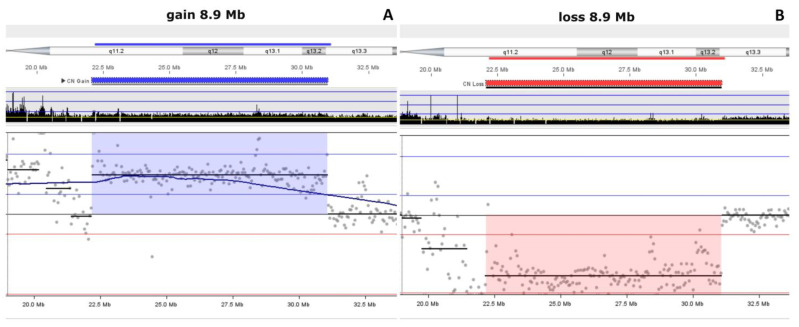
Duplication and deletion of 8.9 Mb on 15q11.2q13.3 identified by LP-WGS. (**A**). Proband duplication characterizing 15q11q13 microduplication syndrome. (**B**) Proband’s sister deletion characterizing Angelman syndrome.

**Figure 2 genes-15-01546-f002:**
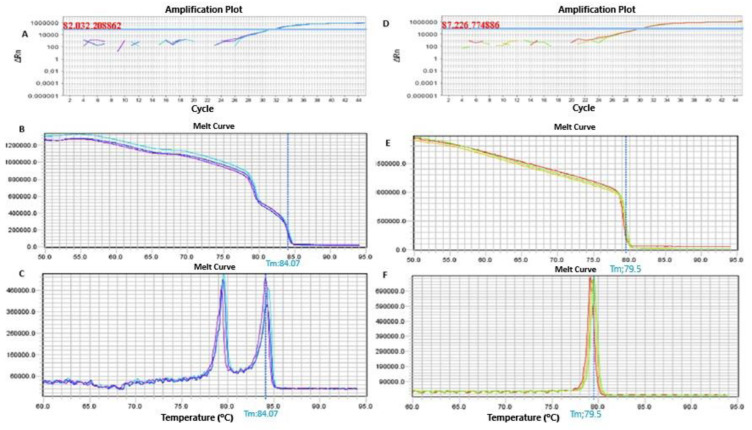
Methylation pattern of proband’s sister and proband, analyzed by MS-HRM. Amplifications plot related to normal (**A**), and Angelman syndrome (**B**). Normalized graphs (**C**,**D**) show initial fluorescence, where all products are double-stranded and bound to the maximum amount of dye. Normal patients present fluorescence drops corresponding to both paternal and maternal alleles (**B**). As the temperature rose, PCR products dissociated, releasing the dye and decreasing the fluorescent signal. The temperature differences between paternal and maternal alleles are attributed to the CpG binding chemistry. Methylated cytosines are nonreactive to bisulfite conversion while nonmethylated cytosines convert to uracil. CpG-rich regions require higher dissociation temperatures. The melting temperature detected for methylated maternal allele was 84.07 °C, and 79.5 °C for the nonmethylated paternal allele. Derivative graphs (**E**,**F**) illustrate the melting peak of each allele. The normal patient in derivative graphs (**E**) presents two peaks corresponding to the unmethylated (79.5 °C) and methylated (84.07 °C) alleles. The absence of maternal allele confirms Angelman syndrome (**D**,**F**). The different colors indicate experimental triplicates.

**Figure 3 genes-15-01546-f003:**
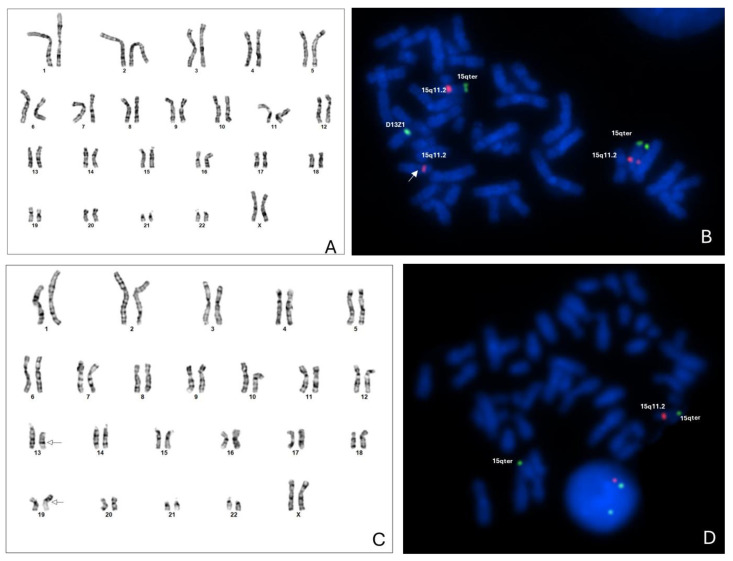
(**A**) Proband GTG analysis—Karyotype: 46,XX. (**B**) The FISH technique using Prader–Willi/Angelman region probe (*SNRPN*) demonstrating three red signals corresponding to 15q11.2 region in the proband. The arrow indicates the third red signal, confirming triplicate of the 15q11.2 region. (**C**) Proband´s sister analysis by GTG banding—Karyotype: 46,XX, t(13;19)(q22;p13.1). (**D**) The FISH analysis using Prader–Willi/Angelman region probe *(SNRPN*) demonstrating deletion of 15q11.2 region in proband’s sister.

**Figure 4 genes-15-01546-f004:**
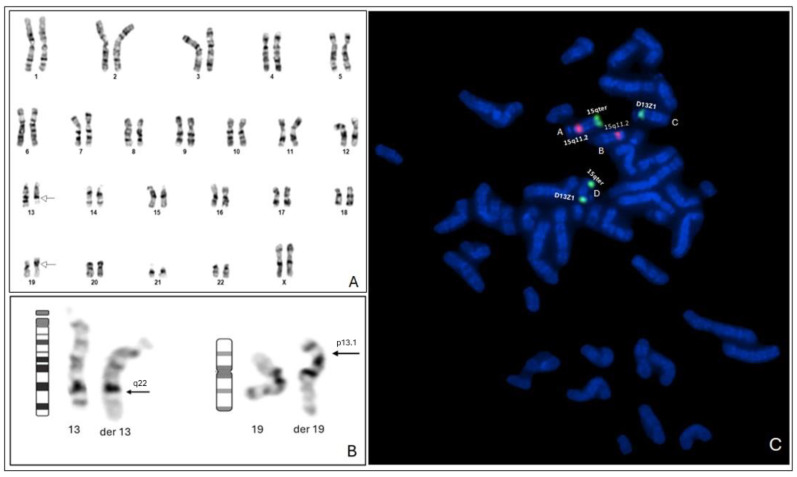
(**A**) Mother’s karyotype by GTG banding—46,XX, t(13;19)(q22;p13.1) (**B**) Partial mother’s karyotype demonstrating translocation t(13;19) with breakpoints (arrows) (**C**) Mother’s FISH analysis using Prader–Willi/Angelman region probe (SNRPN) and chromosome 13 centromeric probe (D13Z1) showing a balanced translocation between the other homologue of chromosome 13 with one of chromosomes 15 (.ish t(13;15)(q12?;q12?) identified two green signals (D13Z1 and 15qter) at the same chromosome (derivative chromosome 13), a normal chromosome 15 (both signals red and green), and a derivative chromosome 15 showing only the red signal (15q11.2). The chromosome marked only with centromeric probe (D13Z1) is the derivative of t(13;19).

**Figure 5 genes-15-01546-f005:**
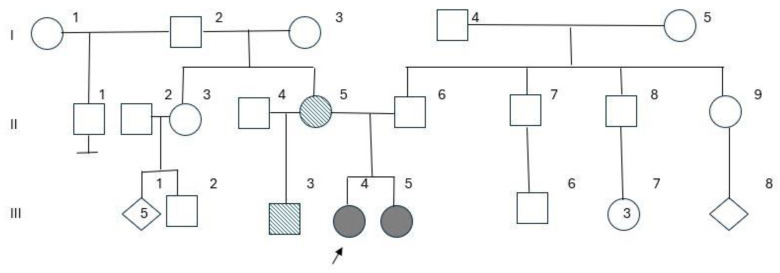
Three-generation pedigree. Shaded symbols denote affected individuals (III-4 = case 1 and III-5 = case 2), and stripe-patterned symbols indicate carriers of balanced translocations. Individuals I-2, I-3, II-3, II-6, and III-2 have a normal karyotype and/or FISH findings, with III-2 clinically diagnosed with Marfan syndrome and a variant of uncertain significance at seq[GRCh38] 15q13.3(31739865_32239864)x3, which is likely benign. The proband is marked with an arrow.

## Data Availability

The datasets generated and analyzed during the current study are not publicly available due to patient data’s confidentiality and ethical aspects but are available from the corresponding author at a reasonable request.

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
