# Peer review of "Clinical and Cytogenetic Impact of Maternal Balanced Double Translocation: A Familial Case of 15q11.2 Microduplication and Microdeletion Syndromes with Genetic Counselling Implications"

_genes, 2024, doi:10.3390/genes15121546_

Round 1
Reviewer 1 Report
Comments and Suggestions for Authors
This manuscript details the cases of a 15q11.2 microduplication and microdeletion syndromes.
Overall, it is well-written, well-presented, and of significant interest to a broad clinical/geneticist readership.
In the title, I would recommend specifying “a familial case of…” to be as clear/specific as possible.
For the Pubmed search, were Boolean operators used? This would be good to include so others can reproduce any analyses, and/or access referenced papers readily.
In Figure 4, would it make sense to indicate with an arrow the proband?
When mentioning databases (e.g. page 7, Genome Browser/DECIPHER), please mention the versions in the text for the greatest specificity possible.
In the first paragraph of the results/discussion section, might it make sense to include an additional sentence or two on the role of SNRPN in neurodevelopment, and a phenotypic link to the observed symptomatology in the siblings? This seems critical to the case report, even if theoretical (e.g. by citing relevant references that have done functional work on SNRPN gene disruptions or other).
In the discussion, while you mention the (very important) need to use multiple sequencing technologies, might it be possible to comment briefly on the increasingly prevalent role of 100x WGS, as well, especially, various long-read sequencing technologies, which are increasingly useful for resolving complex structural variants? It could be worth mentioning the cost/benefit equation relevant to any clinical genetic diagnostic lab. This would be of interest to any clinically inclined reader.
In the discussion – for a clinical prescriptive stance which could add clout to your manuscript - might it make sense to briefly discuss the importance of routine clinical genetic sequencing, e.g. prenatally? This is of great interest in the context of the new era of precision medicine (and could be supported by financial projections).
On a more evolutionary level/broader scale, might you be able to briefly discuss theories behind the emergence of structural variants at certain breakpoints (e.g. in the context of hypermutable regions for example), and/or the theory behind the putative role of structural variants on genome evolution/speciation (e.g. PMID 34284881; PMID 34356100 among others)? This would be of great interest to placing this clinically important work in a broader evolutionary/more philosophical context.
Author Response
Dear Reviewer,
I hope this message finds you well. I would like to express my sincere gratitude for your thoughtful suggestions and comments on my manuscript. Your insights have significantly enhanced the quality, readability, and overall comprehension of the work.
I appreciate the time and effort you dedicated to reviewing my submission, and I am confident that the revisions will contribute positively to the final version of the article.
Thank you once again for your valuable feedback.
Warm regards,
Comment #1: In the title, I would recommend specifying “a familial case of…” to be as clear/specific as possible.
Answer: We change the tittle to:
"Clinical and Cytogenetic Impact of Maternal Balanced Double Translocation: A Familial Case of 15q11.2 Microduplication and Microdeletion Syndromes with Genetic Counseling Implications."
Comment #2: For the Pubmed search, were Boolean operators used? This would be good to include so others can reproduce any analyses, and/or access referenced papers readily.
Answer: We included in the results:
A systematic PubMed search using Boolean operators provided targeted insights into the relationship between 15q11.2 rearrangements and complex chromosomal conditions. For the query "(15q11.2 duplication) AND (complex rearrangements)," 10 results were retrieved, of which only 3 were relevant. The search terms "(15q11.2 duplication) AND (balanced double translocation)" and "(15q11.2 deletion) AND (balanced double translocation)" returned no results, while "(15q11.2 deletion) AND (complex rearrangements)" yielded 6 results, with only 1 being relevant. For "(Angelman syndrome) AND (complex rearrangements)," 17 results were obtained, of which 5 were considered relevant, but no results were found with the combination "(Angelman syndrome) AND (balanced double translocation).
Comment #3: In Figure 4, would it make sense to indicate with an arrow the proband?
Answer:The proband is indicated with an arrow.
Comment #4: When mentioning databases (e.g. page 7, Genome Browser/DECIPHER), please mention the versions in the text for the greatest specificity possible.
Answer: Thank you for your valuable suggestion. In response, we have now included the specific versions of the databases used in our analyses. We referenced the UCSC Genome Browser based on the hg38 (GRCh38) genome assembly and used version v11.28 of the DECIPHER database. This addition ensures the reproducibility and specificity of our findings.
Comment #5: In the first paragraph of the results/discussion section, might it make sense to include an additional sentence or two on the role of SNRPN in neurodevelopment, and a phenotypic link to the observed symptomatology in the siblings? This seems critical to the case report, even if theoretical (e.g. by citing relevant references that have done functional work on SNRPN gene disruptions or other).
Answer: We include:
”The SNRPN gene encodes proteins involved in mRNA processing, vital for various neuronal activities. Located on chromosome 15, the SNRPN locus is essential for epigenetic control, where disruptions contribute to neurodevelopmental syndromes. In PWS, there are distinct cognitive and behavioral manifestations alongside hypothalamic dysregulation, affecting growth and appetite. In AS, which results from maternal deletions or mutations within this region, symptoms include severe developmental delay, motor dysfunction, and seizures. These neurodevelopmental disorders reflect the critical influence of SNRPN on brain development and behavior​ (17, 18).
Furthermore, deletions in the SNORD116 region, associated with SNRPN, result in metabolic and behavioral disruptions in PWS due to appetite and growth dysregulation. Studies on PWS and AS patients, along with mouse models, have shown that genetic disruptions in these areas lead to circadian rhythm gene desynchronization, likely impacting neurodevelopment and behavior (18).”
Comment #6: In the discussion, while you mention the (very important) need to use multiple sequencing technologies, might it be possible to comment briefly on the increasingly prevalent role of 100x WGS, as well, especially, various long-read sequencing technologies, which are increasingly useful for resolving complex structural variants? It could be worth mentioning the cost/benefit equation relevant to any clinical genetic diagnostic lab. This would be of interest to any clinically inclined reader.
Comment #7: In the discussion – for a clinical prescriptive stance which could add clout to your manuscript - might it make sense to briefly discuss the importance of routine clinical genetic sequencing, e.g. prenatally? This is of great interest in the context of the new era of precision medicine (and could be supported by financial projections).
Answer: We include in the discussion:
“Given the complexity of the genetic findings in this family, comprehensive genetic counseling plays a pivotal role in managing the reproductive and familial implications. While the recurrence risk for the translocation or related chromosomal imbalances remains low, the variability in phenotypic expression among offspring highlights the need for careful prenatal screening in future pregnancies. Techniques such as karyotyping and low-pass whole genome sequencing (LP-WGS) may help identify chromosomal abnormalities early, allowing for informed reproductive decisions [19]. Counseling should also include discussions about the potential risks of passing on either balanced translocations or chromosomal imbalances, as even balanced carriers may experience reproductive challenges. Early genetic counseling, combined with appropriate prenatal testing, will aid in providing the family with the necessary tools to anticipate and manage future reproductive outcomes [20].
Moreover, the incorporation of 100x whole genome sequencing (WGS) offers deeper genome coverage, enhancing the detection of single nucleotide variants (SNVs), insertions, deletions, and copy number variations (CNVs). Long-read sequencing technologies, such as those provided by Pacific Biosciences (PacBio) and Oxford Nanopore, enable the sequencing of longer DNA fragments. This capability is particularly advantageous for capturing repetitive regions and large structural variants often missed by traditional approaches. Research indicates that these technologies can significantly improve the accuracy of variant detection and facilitate the assembly of complex genomic regions [21].
As the cost-effectiveness of these sequencing technologies continues to improve, clinical genetic diagnostic labs can adopt WGS as a standard tool. This not only provides comprehensive genomic information to inform clinical decisions and enhance patient outcomes but also allows for the uncovering of previously undiagnosed genetic conditions, enabling tailored treatment strategies. Integrating advanced sequencing technologies into clinical practice not only enhances diagnostic capabilities but also reflects the ongoing evolution of precision medicine, ultimately benefiting patient care and research initiatives [22].”

Reviewer 2 Report
Comments and Suggestions for Authors
The manuscript presented by Vieira and collaborators describes a family in which a karyotype with multiple translocations was identified. One of these rearrangements occurred in the chromosomal region involved in the onset of Angelman/Prader-Willi syndrome. The analysis was performed with various methodologies: classical and molecular cytogenetics, low pass whole genome sequencing, methylation-sensitive high-resolution melting. The results seem to indicate the presence of a karyotype with various anomalies that cannot be detected with just one of the techniques used, but can be detected in their entirety by applying all of them. The authors therefore conclude on the importance of carrying out, during the genetic counseling phase, broad-spectrum cytogenetic investigations, using both classical/molecular cytogenetic techniques and molecular techniques based on PCR.
The manuscript has several weaknesses and critical points that do not allow to be evaluated positively. The presence of a mixed section of "Results and discussion" makes the results difficult to read, since the presentation of the results is the most problematic part that I detected.
In details, these are my criticism and the relative suggestions to the authors to improve the manuscript:
Introduction.
It is very short but can be considered sufficient. The sub-section "case report" I think is not appropriated in this section. I suggest moving the “case report” to the Results section.
Materials and methods.
Section 2.3.
This section is not described exhaustively, such as lacking procedure details (reagents, times, etc.).
Moreover, only one type of probe is indicated, SNRPN/GABRB3, without specifying its features, while the results show ISH with further probes not specified in this section.
Section 2.5.
The primers used (PWS_F and PWS_R) were not indicated, as well as the size of the amplified DNA and the position within the region of interest.
The citation Wojdacz and Hansen 2006 is not present in the bibliography.
The 40 cycles indicated for the PCR are inconsistent with the amplification plot in figure 5, where 45 cycles are indicated.
It is not clear what is meant by “The annealing temperature of 78°C – 83°C was chosen...” perhaps it refers to the observed melting temperatures?
Section 2.6.
What do the two asterisks in the title indicate?
This section is very broad and describes the procedure followed to identify the studies of interest for the presented work: but where are the results? In the "Results and Discussion" section, some sentences present in the "Materials and Methods" are repeated but the results of this bibliographical analysis are not shown. Furthermore, the bibliography is rather limited having cited only 18 publications.
Results and Discussion.
This part is a bit confusing, and it is difficult to follow the flow of the results.
It would be much more useful, for a clearer exposition and a better reading, to separate the section "Results and Discussion" in two different sections: "Results" and "Discussion".
I suggest moving the description of the family (case report) as the first subsection. The family tree (figure 4) should be placed as the first figure and the family members should be indicated, along the text, according to their position in the family tree. In addition, the two sisters (proband and sister) should be clearly indicated as case-1 and case-2, since it is sometimes unclear who the results shown refer to.
Figure 1.
Part A shows a G-banded karyotype. From the caption it seems to have been obtained after FISH. Is this correct? Or was the GTG banding obtained according to standard protocols? As specified in section 2.3.
Why move chromosome 13 next to the two chromosomes 15? Chromosome 13 should remain next to its homologue.
Part B (ISH): some probes not described in the materials and methods are indicated. Three green signals are visible, one of these seem the cen13. And the other two?
The SNRPN probe (red) is detected in three chromosomes, so the duplicate copies are on two different chromosomes?
What does monosomy 13q10 indicate? What band is it?
Figure-2.
Part B: What does the green signal indicate?
Why is an ISH not shown that highlights t(13;19)? A chromosome painting with these two chromosomes would be necessary.
Figure-3.
From which individual was this figure obtained?
Part B: There are two probes with the same green fluorochrome. Here it seems that the centromere of chromosome 13 has not been lost but is located in chromosome 15, contrary to what is shown in figure 1, where a centromere of 13 is no longer visible. It is necessary to perform ISH in which each probe has a different color. And the use of chromosome paintings would give much clearer results.
On page 7 it is indicated that other relatives were analyzed: which individuals? With which methods? With what results?
The results concerning the extensive analysis of the literature is not clear. The obtained result should be better detailed, or this part should be eliminated from the manuscript.
Figure-5.
This figure presents some unclear parts.
Second line: ...AS (D-E) should be ...AS (D-F). Other references also do not correspond with the various parts of the figure and the caption of the entire figure should be revised.
The amplification plots show lines of different colors; it should be specified what they correspond to.
In B two samples with different colors are shown, while in C there seems to be only one sample. In D it is not clear how many samples are present, and two of these are with very high Ct that make the result very weak.
Conclusions.
It is quite limited. It would be useful to expand it and better specify what are the weaknesses and strengths of each of the procedures used and when it is necessary to use them together for better patient management.
In addition, there are minor errors or sentence not clear along the text. Unfortunately, I cannot indicate them precisely since the manuscript lacks line numbers on one side.
Author Response
Dear Reviewer,
I hope this message finds you well. I would like to express my sincere gratitude for your thoughtful suggestions and comments on my manuscript. Your insights have significantly enhanced the quality, readability, and overall comprehension of the work.
I appreciate the time and effort you dedicated to reviewing my submission, and I am confident that the revisions will contribute positively to the final version of the article.
Thank you once again for your valuable feedback.
Warm regards,
Comment #1: Introduction.
It is very short but can be considered sufficient. The sub-section "case report" I think is not appropriated in this section. I suggest moving the “case report” to the Results section.
Answer: We prefer to keep the case report in this section.
Comment #2: Materials and methods.
Section 2.3.
This section is not described exhaustively, such as lacking procedure details (reagents, times, etc.).
Moreover, only one type of probe is indicated, SNRPN/GABRB3, without specifying its features, while the results show ISH with further probes not specified in this section.
Answer: We added the solicitated specifications, including the D13Z1 probe in the text.
Comment #3: Section 2.5.
The primers used (PWS_F and PWS_R) were not indicated, as well as the size of the amplified DNA and the position within the region of interest.
Answer: We included in the text.
Comment #4: The citation Wojdacz and Hansen 2006 is not present in the bibliography.
Answer: We included in the bibliography.
Comment #5: The 40 cycles indicated for the PCR are inconsistent with the amplification plot in figure 5, where 45 cycles are indicated.
Answer: We corrected it in the text.
Comment #6: It is not clear what is meant by “The annealing temperature of 78°C – 83°C was chosen...” perhaps it refers to the observed melting temperatures?
Answer: We switched to: “The amplification products from unmethylated and methylated alleles had a melting temperature of 78°C and 83°C, respectively.”
Comment #7: Section 2.6.
What do the two asterisks in the title indicate?
Answer: This was an editing error and has now been removed
Comment #8: This section is very broad and describes the procedure followed to identify the studies of interest for the presented work: but where are the results? In the "Results and Discussion" section, some sentences present in the "Materials and Methods" are repeated but the results of this bibliographical analysis are not shown. Furthermore, the bibliography is rather limited having cited only 18 publications.
Answer: We included in the results:
A systematic PubMed search using Boolean operators provided targeted insights into the relationship between 15q11.2 rearrangements and complex chromosomal conditions. For the query "(15q11.2 duplication) AND (complex rearrangements)," 10 results were retrieved, of which only 3 were relevant. The search terms "(15q11.2 duplication) AND (balanced double translocation)" and "(15q11.2 deletion) AND (balanced double translocation)" returned no results, while "(15q11.2 deletion) AND (complex rearrangements)" yielded 6 results, with only 1 being relevant. For "(Angelman syndrome) AND (complex rearrangements)," 17 results were obtained, of which 5 were considered relevant, but no results were found with the combination "(Angelman syndrome) AND (balanced double translocation).
Comment #9: Results and Discussion.
This part is a bit confusing, and it is difficult to follow the flow of the results.
It would be much more useful, for a clearer exposition and a better reading, to separate the section "Results and Discussion" in two different sections: "Results" and "Discussion".
Answer: Thank you for your valuable feedback. We have taken your suggestion into consideration and have rewritten the Results and Discussion section. We believe this change will provide a clearer exposition and facilitate a better reading experience for our audience.
Comment #10: I suggest moving the description of the family (case report) as the first subsection. The family tree (figure 4) should be placed as the first figure and the family members should be indicated, along the text, according to their position in the family tree. In addition, the two sisters (proband and sister) should be clearly indicated as case-1 and case-2, since it is sometimes unclear who the results shown refer to.
Answer: Thank you for your valuable feedback. We have taken your suggestion into consideration and have rewritten the Results and Discussion section.
Comment #11: Figure 1.
Part A shows a G-banded karyotype. From the caption it seems to have been obtained after FISH. Is this correct? Or was the GTG banding obtained according to standard protocols? As specified in section 2.3.
Answer: The GTG technique was performed according to the standard protocol described in section 2.3 and was obtained before FISH. However, the karyotype was reinterpreted after FISH.
Why move chromosome 13 next to the two chromosomes 15? Chromosome 13 should remain next to its homologue.
Answer: Since the FISH analysis showed the presence of three red signals, corresponding to the 15q11.2 region and only one green signal, corresponding to the 13 centromere, we understand that there is only one chromosome 13, two chromosomes 15 and one derivative chromosome 15, since according to ISCN 2020, the derivative chromosome is the one that keeps the centromere intact.
Part B (ISH): some probes not described in the materials and methods are indicated. Three green signals are visible, one of these seem the cen13. And the other two?
Answer: This probe was included in section 2.3. We identified all the probes signals in Figure2-B as suggested.
The SNRPN probe (red) is detected in three chromosomes, so the duplicate copies are on two different chromosomes?
Answer: Yes, this is the consequence of maternal balanced translocation between 13;15 chromosomes, that was only identified by FISH (cryptic rearrangement).
What does monosomy 13q10 indicate? What band is it?
Answer: Indicate that there´s only one centromeric region from chromosome 13. This is centromeric region.
Comment #12: Figure-2.
Part B: What does the green signal indicate?
Answer: Indicate the centromeric region of chromosomo 13. We included this information in the figure 2.
Why is an ISH not shown that highlights t(13;19)? A chromosome painting with these two chromosomes would be necessary.
Answer: The 13;19 translocation is clearly visible by GTG banding analysis and does not require confirmation by FISH.
Comment #13: Figure-3.
From which individual was this figure obtained?
Answer: The figures 3A and 3B refers to the sample of the proband's younger sister, who has an absence of the 15q11.2 region, identified by LP-WGS and FISH. In 3C we show the duplication (which occurred in the proband) and the deletion (which occurred in the sister).
Part B: There are two probes with the same green fluorochrome. Here it seems that the centromere of chromosome 13 has not been lost but is located in chromosome 15, contrary to what is shown in figure 1, where a centromere of 13 is no longer visible. It is necessary to perform ISH in which each probe has a different color. And the use of chromosome paintings would give much clearer results.
Answer: The centromere of chromosome 13 was not lost in the mother but was translocated to chromosome 15. However, the segregation of the chromosomes during maternal meiosis resulted in the absence of the chromosome 13 derivative in the daughter, leading to monosomy of the 13q10 region.
In this case, interpretation was possible, even using the same-colored probes, since we already had the previous result from the analysis of the mother's material.
Comment #14: On page 7 it is indicated that other relatives were analyzed: which individuals? With which methods? With what results?
Answer: We added this information in the text “Family members, including the father, brother, maternal aunt and grandparents were evaluated by GTG banding analysis and FISH technique, presenting normal results, suggesting this cryptic translocation arose de novo in the mother.”
The results concerning the extensive analysis of the literature is not clear. The obtained result should be better detailed, or this part should be eliminated from the manuscript.
Answer: We improved this discussion bellow the Figure 4.
Comment #15: Figure-5.
This figure presents some unclear parts.
Second line: ...AS (D-E) should be ...AS (D-F). Other references also do not correspond with the various parts of the figure and the caption of the entire figure should be revised.
The amplification plots show lines of different colors; it should be specified what they correspond to.
In B two samples with different colors are shown, while in C there seems to be only one sample. In D it is not clear how many samples are present, and two of these are with very high Ct that make the result very weak.
Answer: We improve the figure and corrected the legend
Comment #16: Conclusions.
It is quite limited. It would be useful to expand it and better specify what are the weaknesses and strengths of each of the procedures used and when it is necessary to use them together for better patient management.
Answer: We have rewritten the conclusions:
This case exemplifies the significant impact that complex chromosomal rearrangements can have on neurodevelopment, highlighting the critical need for a multidisciplinary diagnostic approach when investigating such anomalies. The use of complementary cytogenetic and molecular techniques enabled the identification of cryptic balanced translocations, duplications, and deletions that significantly influence clinical outcomes. Specifically, the presence of 15q11.2q13.3 duplication and deletion in siblings underscores the potential for balanced translocations to result in diverse and severe neurodevelopmental disorders. These findings underscore the importance of comprehensive genetic counseling for families affected by rare chromosomal anomalies. Such counseling should address both reproductive implications and the lifelong management of neurodevelopmental and physical challenges that may arise due to these genetic rearrangements.
Comment #17: In addition, there are minor errors or sentence not clear along the text. Unfortunately, I cannot indicate them precisely since the manuscript lacks line numbers on one side.
Answer: Thank you for your constructive feedback regarding the clarity and writing errors in the manuscript. We have thoroughly re-evaluated the text for any minor errors and unclear sentences. To address your concern about the lack of line numbers, we have added line numbering to the manuscript for easier reference in future reviews. We appreciate your understanding and the opportunity to improve our work.

Round 2
Reviewer 2 Report
Comments and Suggestions for Authors
There are several aspects that the authors have not considered and that I believe are important for the valorization of their results:
1. Case report: it cannot be included in the introduction. If it is a new description, it should be included in the results. If it has already been described it should be included in the introduction with the relative citation.
2. The materials and methods continue to be inadequate for a scientific article. The procedures must be described in full, it is not enough to write “standard procedure". Also, because it is not a standard procedure to first do a GTG banding and then do a FISH on the same preparations. Why? If it is so important authors should show a metaphase with GTG banding and the same metaphase with FISH. This section must therefore be better detailed, obviously with due citations. The probes used for FISH must be well detailed: name of the probe and supplier company.
3. The size of the MS-HRM amplified product and its genomic position were not indicated.
4. In figure 2 is shown the probe 15q11.2 in a chromosome that however cannot be said, with this FISH, to be a chromosome 19.
5. Monosomy 13q10: It is very strange that a centromere is indicated as band q10 (why not p10?). Maybe in the ISCN 2020 system the centromeres are indicated in this way?
6. Figure 3: the name of the probe used in FISH is missing.
7. Figure 4: it is very difficult to recognize a translocation that includes chromosome 19 with short chromosomes. Don't you have a metaphase at higher resolution?
8. Figure 5. In A and B I think that a graph with linear curves renders better than a log curve. The title of the other graphs (melting curve) disappeared.
9. I always suggest separating the results from the conclusions, to make the description of the results clearer.
Author Response
Dear Reviewer,
I hope this message finds you well. I would like to express my sincere gratitude for your thoughtful suggestions and comments on my manuscript.
The format of the article was modified, according to your suggestions, by including the case report in the results and separating the results from the discussion, leading to a change in the order of the figures.
Thank you once again for your valuable feedback.
Warm regards,
Elenice Bastos and authors
Comment #1: 1. Case report: it cannot be included in the introduction. If it is a new description, it should be included in the results. If it has already been described it should be included in the introduction with the relative citation.
Answer: It was included in the results.
Comment #2: The materials and methods continue to be inadequate for a scientific article. The procedures must be described in full, it is not enough to write “standard procedure". Also, because it is not a standard procedure to first do a GTG banding and then do a FISH on the same preparations. Why? If it is so important authors should show a metaphase with GTG banding and the same metaphase with FISH. This section must therefore be better detailed, obviously with due citations. The probes used for FISH must be well detailed: name of the probe and supplier company
Answer: We included the full description, as suggested
Comment #3: The size of the MS-HRM amplified product and its genomic position were not indicated.
Answer: We included the sentence: “The amplicons size of methylated maternal allele and the nonmethylated paternal allele is 187bp” - line 161
Comment #4: In figure 2 is shown the probe 15q11.2 in a chromosome that however cannot be said, with this FISH, to be a chromosome 19
Answer: Figure 2 became figure 3. This figure has been modified to include the GTG karyotypes of both sisters for a better understanding. The SNRPN probe (15q11.2) does not mark chromosome 19, since this chromosome is not involved in the t(13;15) translocation identified by FISH.
Comment #5: Monosomy 13q10: It is very strange that a centromere is indicated as band q10 (why not p10?). Maybe in the ISCN 2020 system the centromeres are indicated in this way
Answer: In fact, according to ISCN 2020, the band involving the centromeric region should be described as p11.1-q11.1
Comment #6: Figure 3: the name of the probe used in FISH is missing.
Answer: Figure 3 became Figure 1, showing only the LP-WGS results.
Comment #7: Figure 4: it is very difficult to recognize a translocation that includes chromosome 19 with short chromosomes. Don't you have a metaphase at higher resolution?.
Answer: Figure 4 has been modified. We have included a new image of the mother's partial karyotype in GTG banding, which better demonstrates the t(13;19) translocation.
The translocation identified by FISH occurred between one of chromosomes 15 and the homologous chromosome 13 not involved in the t(13;19) translocation visualized by GTG banding.
These are two independent and concomitant balanced translocations.
We currently don't have a painting probe for chromosome 19.
Comment #8: Figure 5. In A and B I think that a graph with linear curves renders better than a log curve. The title of the other graphs (melting curve) disappeared.
Answer: The default is to use the logarithmic curve. We corrected the title of the other graphs and included the term fusion curve.
Comment #9: I always suggest separating the results from the conclusions, to make the description of the results clearer.
Answer: We accepted your suggestion and separated the results of the discussion.

Round 3
Reviewer 2 Report
Comments and Suggestions for Authors
The authors have revised the manuscript following several of the suggestions provided. However, there are some aspects that should be improved or clarified:
- the metaphases with GTG banding (Figures 3A, 3C, 4A) do not seem to be the same as the metaphases with FISH shown on the right (Figures 3B, 3D, 4C). Since this seems to be a relevant aspect in the analysis, the GTG and FISH metaphases should match.
- the karyotype 4A (mother) is the same as the karyotype 3C (proband's sister). Obviously one of the two must be replaced.
- the FISH images seem to me to have a black background that is not real. The original images should be put.
- it is not clear to me why the pedigree has been moved to the discussion. It would have been useful for the reader to have it immediately visible at the beginning of the results. But if the authors think this position is better for me it is fine.
- there are several typos throughout the text. For example, in line 268 there are two: Proband´s sirter [sister] analysis by GTG banding – Karyotype: 46.XX, t(13;19)(q22;p13.1). [46,XX, ......].
- Throughout the text, the karyotype 46,XX is sometimes indicated with a dot and sometimes with a comma (46.XX or 46,XX).
Author Response
Dear Reviewer,
I hope this message finds you well. I would like to express my sincere gratitude for your thoughtful suggestions and comments on my manuscript.
We made all corrections suggested and try to clarify some solicitated aspects.
Thank you once again for your valuable feedback.
Warm regards,
Elenice Bastos and authors
Comment 1:
- the metaphases with GTG banding (Figures 3A, 3C, 4A) do not seem to be the same as the metaphases with FISH shown on the right (Figures 3B, 3D, 4C). Since this seems to be a relevant aspect in the analysis, the GTG and FISH metaphases should match.
Answer: In fact, the metaphases are not the same. The GTG banding and FISH techniques were carried out on the same samples from each patient, but on different slides.
They were independent analyses whose results complement each other.
The G-banding technique followed by FISH, on the same slide, was not considered essential for the unequivocal identification of independent translocations.
Comment 2
- the karyotype 4A (mother) is the same as the karyotype 3C (proband's sister). Obviously one of the two must be replaced.
Answer: Thank you for your observation. It was a mistake. We corretected it including a new image from mother’s proband in figure 4A
Comment 3
- the FISH images seem to me to have a black background that is not real. The original images should be put.
Answer: All images are real. It isn´t black but a “dark blue”
Comment 4
- it is not clear to me why the pedigree has been moved to the discussion. It would have been useful for the reader to have it immediately visible at the beginning of the results. But if the authors think this position is better for me it is fine.
Answer: We really think that in this position it's easier to correlate the pedigree with the discussion, so we prefer to keep it.
Comment 5
- there are several typos throughout the text. For example, in line 268 there are two: Proband´s sirter [sister] analysis by GTG banding – Karyotype: 46.XX, t(13;19)(q22;p13.1). [46,XX, ......].
Answer: It was corrected.
Comments 6
- Throughout the text, the karyotype 46,XX is sometimes indicated with a dot and sometimes with a comma (46.XX or 46,XX).
Answer: It was corrected.